# RetGK: Graph Kernels based on Return Probabilities of Random Walks

**Zhen Zhang, Mianzhi Wang, Yijian Xiang, Yan Huang, and Arye Nehorai**
Department of Electrical and Systems Engineering
Washington University in St. Louis
St. Louis, MO 63130
{zhen.zhang, mianzhi.wang, yijian.xiang, yanhuang640, nehorai}@wustl.edu

## Abstract

Graph-structured data arise in wide applications, such as computer vision, bioinformatics, and social networks. Quantifying similarities among graphs is a fundamental problem. In this paper, we develop a framework for computing graph kernels, based on return probabilities of random walks. The advantages of our proposed kernels are that they can effectively exploit various node attributes, while being scalable to large datasets. We conduct extensive graph classification experiments to evaluate our graph kernels. The experimental results show that our graph kernels significantly outperform existing state-of-the-art approaches in both accuracy and computational efficiency.

## 1 Introduction

Structured data modeled as graphs arise in many application domains, such as computer vision, bioinformatics, and social network mining. One interesting problem for graph-type data is quantifying their similarities based on the connectivity structure and attribute information. Graph kernels, which are positive definite functions on graphs, are powerful similarity measures, in the sense that they make various kernel-based learning algorithms, for example, clustering, classification, and regression, applicable to structured data. For instance, it is possible to classify proteins by predicting whether a given protein is an enzyme or not.

There are several technical challenges in developing effective graph kernels: (i) When designing graph kernels, one might come across the graph isomorphism problem, a well-known NP problem. The kernels should satisfy the isomorphism-invariant property, while being informative on the topological structure difference. (ii) Graphs are usually coupled with multiple types of node attributes, e.g., discrete[1] or continuous attributes. For example, a chemical compound may have both discrete and continuous attributes, which respectively describe the type and position of atoms. A crucial problem is how to integrate the graph structure and node attribute information in graph kernels. (iii) In some applications, e.g., social networks, graphs tend to be very large, with thousands or even millions of nodes, which requires strongly scalable graph kernels.

In this work, we propose novel methods to overcome these challenges. We revisit the concept of random walks, introducing a new node structural role descriptor, the return probability feature (RPF). We rigorously show that the RPF is isomorphism-invariant and encodes very rich connectivity information. Moreover, RPF allows us to consider attributed and nonattributed graphs in a unified framework. With the RPF, we can embed (non-)attributed graphs into a Hilbert space. After that, we naturally obtain our return probability-based graph kernels ("$\mathrm{RetGK}$" for short). Combining with the approximate feature maps technique, we represent each graph with a multi-dimensional tensor and design a family of computationally efficient graphs kernels.

**Related work.** There are various graph kernels, many of which explore the R-convolutional framework [12]. The key idea is decomposing a whole graph into small substructures and building graph kernels based on the similarities among these components. Such kernels differ from each other in the way they decompose graphs. For example, graphlet kernels [26] are based on small subgraphs up to a fixed size. Weisfeiler-Lehman graph kernels [25] and tree-based kernels [6] are developed with subtree patterns. Shortest path kernels [1] are derived by comparing the paths between graphs. Still other graph kernels, such as [30] and [10], are developed by counting the number of common random walks on direct product graphs. Recently, subgraph matching kernels [18] and graph invariant kernels [22] were proposed for handling continuous attributes. However, all the above R-convolution based graph kernels suffer from a drawback. As pointed out in [32], increasing the size of substructures will largely decrease the probability that two graphs contain similar substructures, which usually results in the "diagonal dominance issue" [14]. Our return probability based kernels are significantly different from the above ones. We measure the similarity between two graphs by directly comparing their node structural role distributions, avoiding substructures decomposition.

More recently, new methods have been proposed for comparing graphs, which is done by quantifying the dissimilarity between the distributions of pairwise distances between nodes. [24] uses the shortest path distance, and [29] uses the diffusion distance. However, these methods can be applied only to non-attributed (unlabeled) graphs, which largely limits their applications in the real world.

**Organization**. In Section 2, we introduce the necessary background, including graph concepts and tensor algebra. In Section 3, we discuss the favorable properties of and computational methods for RPF. In Section 4, we present the Hilbert space embedding of graphs, and develop the corresponding graph kernels. In Section 5, we show the tensor representation of graphs, and derive computational efficient graph kernels. In Section 6, we report the experimental results on 21 benchmark datasets. In the supplementary material, we provide proofs of all mathematical results in the paper.

## 2  Background

### 2.1  Graph concepts

An undirect graph $G$ consists of a set of nodes $V_G = \{v_1, v_2, ..., v_n\}$ and a set of edges $E_G \subseteq V_G \times V_G$. Each edge $(v_i, v_j)$ is assigned with a positive value $w_{ij}$ describing the connection strength between $v_i$ and $v_j$. For an unweighted graph, all the edge weights are set to be one, i.e., $w_{ij} = 1, \forall (v_i, v_j) \in E_G$. Two graphs $G$ and $H$ are isomorphic if there exists a permutation map $\tau : V_G \rightarrow V_H$, such that $\forall (v_i, v_j) \in E_G$, $\big(\tau(v_i), \tau(v_j)\big) \in E_H$, and the corresponding edge weights are preserved.

The adjacent matrix $\boldsymbol{A}_G$ is an $n \times n$ symmetric matrix with $\boldsymbol{A}_G(i, j) = w_{ij}$. The degree matrix $\boldsymbol{D}_G$ is diagonal matrix whose diagonal terms are $\boldsymbol{D}_G(i, i) = \sum_{(v_i, v_j) \in E_G} w_{ij}$. The volume of $G$ is the summation of all node degrees, i.e., $\text{Vol}_G = \sum_{i=1}^{n} \boldsymbol{D}_G(i, i)$. An $S$-step walk starting from node $v_0$ is a sequence of nodes $\{v_0, v_1, v_2, ..., v_S\}$, with $(v_s, v_{s+1}) \in E_G, 0 \leq s \leq S - 1$. A random walk on $G$ is a Markov chain $(X_0, X_1, X_2, ...)$, whose transition probabilities are

$$\Pr(X_{i+1} = v_{i+1}|X_i = v_i, ..., X_0 = v_0) = \Pr(X_{i+1} = v_{i+1}|X_i = v_i) = \frac{w_{ij}}{\boldsymbol{D}_G(i, i)}, \quad (1)$$

which induces the transition probability matrix $\boldsymbol{P}_G = \boldsymbol{D}_G^{-1} \boldsymbol{A}_G$. More generally, $\boldsymbol{P}_G^s$ is the $s$-step transition matrix, where $\boldsymbol{P}_G^s(i, j)$ is the transition probability in $s$ steps from node $v_i$ to $v_j$.

In our paper, we also consider the case that nodes are associated with multiple attributes. Let $\mathcal{A}$ denote a attribute domain. Typically, $\mathcal{A}$ can be a alphabet set or a subset of a Euclidean space, which corresponds to discrete attributes and continuous attributes, respectively.

### 2.2  Tensor algebra

A tensor [17] is a multidimensional array, which has multiple indices.[2] We use $\mathbb{R}^{I_1 \times I_2 \times ... \times I_N}$ to denote the set of tensors of order $N$ with dimension $(I_1, I_2, ..., I_N)$. If $U \in \mathbb{R}^{I_1 \times I_2 \times ... \times I_N}$, then $U_{i_1 i_2, ..., i_N} \in \mathbb{R}$, where $1 \leq i_1 \leq I_1, ..., 1 \leq i_N \leq I_N$.

The inner product between tensors $U, V \in \mathbb{R}^{I_1 \times I_2 \times ... \times I_N}$ is defined such that

$$\langle U, V \rangle_{\mathcal{T}} = \text{vec}(U)^T \text{vec}(V) = \sum_{i_1=1}^{I_1} \sum_{i_2=1}^{I_2} ... \sum_{i_N=1}^{I_N} U_{i_1 i_2,...,i_N} V_{i_1 i_2,...,i_N}. \tag{2}$$

A rank-one tensor $W \in \mathbb{R}^{I_1 \times I_2 \times ... \times I_N}$ is the tensor (outer) product of $N$ vectors, i.e., $W = \vec{\boldsymbol{w}}^{(1)} \circ \vec{\boldsymbol{w}}^{(2)} \circ ... \circ \vec{\boldsymbol{w}}^{(N)}$, $W_{i_1 i_2,...,i_N} = \vec{\boldsymbol{w}}_{i_1}^{(1)} \vec{\boldsymbol{w}}_{i_2}^{(2)} ... \vec{\boldsymbol{w}}_{i_N}^{(N)}$.

# 3 Return probabilities of random walks

Given a graph $G$, as we can see from (1), the transition probability matrix, $\boldsymbol{P}_G$, encodes all the connectivity information, which leads to a natural intuition: We can compare two graphs by quantifying the difference between their transition probability matrices. However, big technical difficulties exist, since the sizes of two matrices are not necessarily the same, and their rows or columns do not correspond in most cases.

To tackle the above issues, we make use of the $S$-step return probabilities of random walks on $G$. To do this, we assign each node $v_i \in V_G$ an $S$-dimensional feature called "return probability feature" ("RPF" for short), which describes the "structural role" of $v_i$, i.e.,

$$\vec{\boldsymbol{p}}_i = [\boldsymbol{P}_G^1(i,i), \boldsymbol{P}_G^2(i,i), ..., \boldsymbol{P}_G^S(i,i)]^T, \tag{3}$$

where $\boldsymbol{P}_G^s(i,i)$, $s = 1, 2, ..., S$, is the return probability of a $s$-step random walk starting from $v_i$. Now each graph is represented by a set of feature vectors in $\mathbb{R}^S$: $\text{RPF}_G^S = \{\vec{\boldsymbol{p}}_1, \vec{\boldsymbol{p}}_2, ..., \vec{\boldsymbol{p}}_n\}$. The RPF has three nice properties: isomorphism-invariance, multi-resolution, and informativeness.

## 3.1 The properties of RPF

**Isomorphism-invariance.** The isomorphism-invariance property of return probability features is summarized in the following proposition.

**Proposition 1.** *Let $G$ and $H$ be two isomorphic graphs of $n$ nodes, and let $\tau : \{1, 2, ..., n\} \to \{1, 2, ..., n\}$ be the corresponding isomorphism. Then,*

$$\forall v_i \in V_G, \; s = 1, 2, ..., \infty, \; \boldsymbol{P}_G^s(i,i) = \boldsymbol{P}_H^s(\tau(i), \tau(i)). \tag{4}$$

Clearly, isomorphic graphs have the same set of RPF, i.e., $\text{RPF}_G^S = \text{RPF}_H^S$, $\forall S = 1, 2, ..., \infty$. Such a property can be used to check graph isomorphism, i.e., if $\exists S$, s.t. $\text{RPF}_G^S \neq \text{RPF}_H^S$, then $G$ and $H$ are not isomorphic. Moreover, Proposition 1 allows us to directly compare the structural role of any two nodes in different graphs, without considering the matching problems.

**Multi-resolution.** RPF characterizes the "structural role" of nodes with multi-resolutions. Roughly speaking, $\boldsymbol{P}_G^s(i,i)$ reflects the interaction between node $v_i$ and the subgraph involving $v_i$. With an increase in $s$, the subgraph becomes larger. We use a toy example to illustrate our idea. Fig. 1(a) presents an unweighted graph $G$, and $C_1$, $C_2$, and $C_3$ are three center nodes in $G$, which play different structural roles. In Fig. 1(b), we plot their $s$-step return probabilities, $s = 1, 2, ..., 200$. $C_1, C_2$, and $C_3$ have the same degree, as do their neighbors. Thus their first two return probabilities are the same. Since $C_1$ and $C_2$ share the similar neighbourhoods at larger scales, their return probability values are close until the eighth step. Because $C_3$ plays a very different structural role from $C_1$ and $C_2$, its return probabilities values deviate from those of $C_1$ and $C_2$ in early steps.

In addition, as shown in Fig. 1(b), when the random walk step $s$ approaches infinity, the return probability $\boldsymbol{P}_G^s(i,i)$ will not change much and will converge to a certain value, which is known as the stationary probability in Markov chain theory [5]. Therefore, if $s$ is already sufficiently large, we gain very little new information from the RPF by increasing $s$.

**Informativeness.** The RPF provides very rich information on the graph structure, in the sense that if two graphs has the same RPF sets, they share very similar spectral properties.

**Theorem 1.** *Let $G$ and $H$ be two connected graphs of the same size $n$ and volume Vol, and let $\boldsymbol{P}_G$ and $\boldsymbol{P}_H$ be the corresponding transition probability matrices. Let $\{(\lambda_k, \vec{\psi}_k)\}_{k=1}^n$ and $\{(\mu_k, \vec{\varphi}_k)\}_{k=1}^n$ be eigenpairs of $\boldsymbol{P}_G$ and $\boldsymbol{P}_H$, respectively. Let $\tau : \{1, 2, ..., n\} \to \{1, 2, ..., n\}$ be a permutation map. If $\boldsymbol{P}_G^s(i,i) = \boldsymbol{P}_H^s(\tau(i), \tau(i)), \forall v_i \in V_G, \forall s = 1, 2, ..., n$, i.e., $\text{RPF}_G^n = \text{RPF}_H^n$, then,*

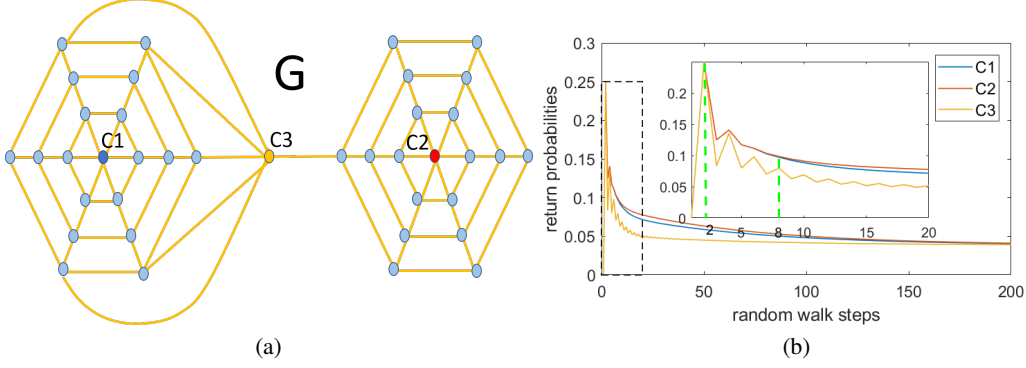

(a)                                            (b)

Figure 1: (a) Toy Graph $G$; (b) The $s$-step return probability of the nodes $C_1$, $C_2$ and $C_3$ in the toy graph, $s = 1, 2, ..., 200$. The nested figure is a close-up view of the rectangular region.

1. $\mathrm{RPF}_G^S = \mathrm{RPF}_H^S$, $\forall S = n + 1, n + 2, ..., \infty$;

2. $\{\lambda_1, \lambda_2, ..., \lambda_n\} = \{\mu_1, \mu_2, ..., \mu_n\}$;

3. *If the eigenvalues sorted by their magnitudes satisfy:* $|\lambda_1| > |\lambda_2| > ... > |\lambda_m| > 0$, $|\lambda_{m+1}| = ... = |\lambda_n| = 0$, *then we have that* $|\vec{\psi}_k(i)| = |\vec{\varphi}_k(\tau(i))|$, $\forall v_i \in V_G$, $\forall k = 1, 2, ..., m$.

The first conclusion states that the graph structure information contained in $\mathrm{RPF}_G^n$ and $\mathrm{RPF}_G^S$, $S \geq n$ are the same, coinciding with our previous discussions on RPF with large random walk steps. The second and third conclusions bridge the RPF with spectral representations of graphs [4], which contains almost all graph structure information.

**Relation to eigenvector embeddings (EE).** One popular way of embedding graph nodes in a Euclidean space uses the eigenvectors of Laplacian or adjacent matrices as the coordinates. In [21], a class of graph kernels are developed based on the eigenvector embeddings. From Theorem 1, we see that both RPF and EE encode the spectral information of graphs. However, our RPF has several advantages over EE. (i) The eigenvector embeddings reflect the closeness among nodes in the same graph, which makes it difficult to compare node across graphs. (ii) The EE representations, which are computed up to a change in sign (or more generally, orthonormal transformation in the eigenspace), may not be invariant under graph isomorphisms. A counterexample is shown in Fig. 2. G and G' are two isomorphic graphs, we visualize their first three-dimensional embeddings with RPF and EE [3]. It can be seen that RPFs are invariant while eigenvectors are not. (iii) The eigenvector embeddings are unstable. The perturbation theory says that two eigenvectors may switch if their eigenvalues are close.

## 3.2 The computation of RPF

Given a graph $G$, the brute-force computation of $\mathrm{RPF}_G^S$ requires $(S-1) \times n \times n$ matrix multiplication of $\boldsymbol{P}_G$. Therefore, the time complexity is $(S-1)n^3$, which is quite high when $S$ is large.

Since only the diagonal terms of transition matrices are needed, we have efficient techniques. Write

$$\boldsymbol{P}_G = \boldsymbol{D}_G^{-1} \boldsymbol{A}_G = \boldsymbol{D}_G^{-\frac{1}{2}} (\boldsymbol{D}_G^{-\frac{1}{2}} \boldsymbol{A}_G \boldsymbol{D}_G^{-\frac{1}{2}}) \boldsymbol{D}_G^{\frac{1}{2}} = \boldsymbol{D}_G^{-\frac{1}{2}} \boldsymbol{B}_G \boldsymbol{D}_G^{\frac{1}{2}}, \tag{5}$$

where $\boldsymbol{B}_G = \boldsymbol{D}_G^{-\frac{1}{2}} \boldsymbol{A}_G \boldsymbol{D}_G^{-\frac{1}{2}}$ is a symmetric matrix. Then $\boldsymbol{P}_G^s = \boldsymbol{D}_G^{-\frac{1}{2}} \boldsymbol{B}_G^s \boldsymbol{D}_G^{\frac{1}{2}}$. Let $\{(\lambda_k, \vec{\boldsymbol{u}}_k)\}_{k=1}^n$ be the eigenpairs of $\boldsymbol{B}_G$, i.e., $\boldsymbol{B}_G = \sum_{k=1}^n \lambda_k \vec{\boldsymbol{u}}_k \vec{\boldsymbol{u}}_k^T$. Then the return probabilities are

$$\boldsymbol{P}_G^s(i, i) = \boldsymbol{B}_G^s(i, i) = \sum_{k=1}^n \lambda_k^s [\vec{\boldsymbol{u}}_k(i)]^2, \forall v_i \in V_G, \forall s = 1, 2, ..., S. \tag{6}$$

Let $\boldsymbol{U} = [\vec{\boldsymbol{u}}_1, \vec{\boldsymbol{u}}_2, ..., \vec{\boldsymbol{u}}_n]$, let $\boldsymbol{V} = \boldsymbol{U} \odot \boldsymbol{U}$, where $\odot$ denotes Hadamard product, and let $\vec{\boldsymbol{\Lambda}}_s = [\lambda_1^s, \lambda_2^s, ..., \lambda_n^s]^T$. Then we can obtain all nodes' $s$-step return probabilities in the vector $\boldsymbol{V}\vec{\boldsymbol{\Lambda}}_s$. The

eigen-decomposition of $\boldsymbol{B}_G$ requires time $O(n^3)$. Computing $\boldsymbol{V}$ or $\boldsymbol{V}\vec{\boldsymbol{\Lambda}}_s$, $\forall s = 1, 2, ..., S$, takes time $O(n^2)$. So the total time complexity of the above computational method is $O\left(n^3 + (S+1)n^2\right)$.

### 3.2.1 Monte Carlo simulation method

If the graph node number, $n$, is large, i.e., $n > 10^5$, the eigendecomposition of an $n \times n$ matrix is relatively time-consuming. To make RPF scalable to large graphs, we use the Monte Carlo method to simulate random walks. Given a graph $G$, for each node $v_i \in V_G$, we can simulate a random walk of length $S$ based on the transition probability matrix $\boldsymbol{P}_G$. We repeat the above procedure $M$ times, obtaining $M$ sequences of random walks. For each step $s = 1, 2, ..., S$, we use the relative frequency of returning to the starting point as the estimation of the corresponding $s$-step return probability. The random walk simulation is parallelizable and can be implemented efficiently, characteristics of which both contribute to the scalability of RPF.

## 4 Hilbert space embeddings of graphs

In this section, we introduce the Hilbert space embeddings of graphs, based on the RPF. With such Hilbert space embeddings, we can naturally obtain the corresponding graph kernels.

As discussed in Section 3, the structural role of each node $v_i$ can be characterized by an $S-$dimensional return probability vector $\vec{\boldsymbol{p}}_i$ (see 3), and thus a nonattributed graph can be represented by the set $\mathrm{RPF}_G^S = \{\vec{\boldsymbol{p}}_i\}_{i=1}^n$. Since the isomorphism-invariance property allows direct comparison of nodes' structural roles across different graphs, we can view the RPF as a special type of attribute, namely, "the structural role attribute" (whose domain is denoted as $\mathcal{A}_0$), associated with nodes. Clearly, $\mathcal{A}_0 = \mathbb{R}^S$.

The nodes of attributed graphs usually have other types of attributes, which are obtained by physical measurements. Let $\mathcal{A}_1, \mathcal{A}_2, ..., \mathcal{A}_L$ be their attribute domains. When combined with RPF, an attributed graph can be represented by the set $\{(\vec{\boldsymbol{p}}_i, a_i^1, ..., a_i^L)\}_{i=1}^n \subseteq \mathcal{A}_0 \times \mathcal{A}_1 \times ... \times \mathcal{A}_L$ (denoted as $\times_{l=0}^L \mathcal{A}_l$). Such a representation allows us to consider both attributed and nonattributed graphs in a unified framework, since if $L = 0$, the above set just degenerates to the nonattributed case. The set representation forms an empirical distribution $\mu = \frac{1}{n} \sum_{i=1}^{n} \delta_{(\vec{\boldsymbol{p}}_i, a_i^1, ..., a_i^L)}$ on $\mathcal{A} = \times_{l=0}^L \mathcal{A}_l$, which can be embedded into a reproducing kernel Hilbert space (RKHS) by kernel mean embedding [11].

Let $k_l$, $l = 0, 1, ..., L$ be a kernel on $\mathcal{A}_l$. Let $\mathcal{H}_l$ and $\phi_l$ be the corresponding RKHS and implicit feature map, respectively. Then we can define a kernel on $\mathcal{A}$ through the tensor product of kernels [28], i.e., $k = \otimes_{l=0}^L k_l$, $k\left[(\vec{\boldsymbol{p}}, a^1, a^2, ..., a^L), (\vec{\boldsymbol{q}}, b^1, b^2, ..., b^L)\right] = k_0(\vec{\boldsymbol{p}}, \vec{\boldsymbol{q}}) \prod_{l=1}^L k_l(a^l, b^l)$. Its associated RKHS, $\mathcal{H}$, is the tensor product space generated by $\mathcal{H}_l$, i.e., $\mathcal{H} = \otimes_{l=0}^L \mathcal{H}_l$. Let $\phi : \mathcal{A} \to \mathcal{H}$ be the

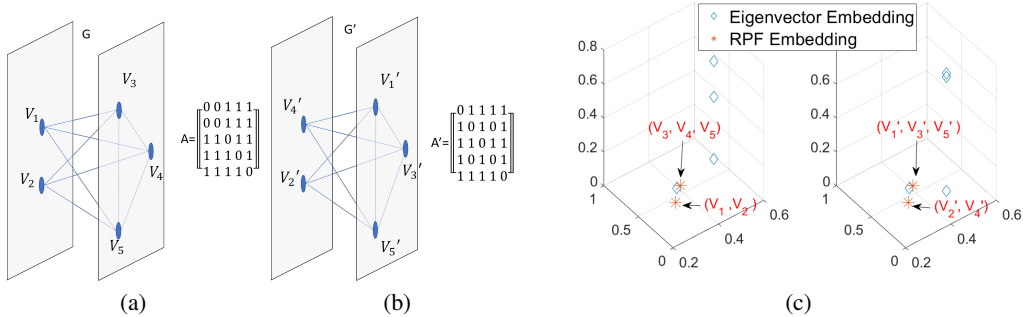

Figure 2: Toy graph G and its adjacent matrix; (b) Toy graph G' and its adjacent matrix; (c) 3-D eigenvector and RPF embeddings of nodes in G and G', respectively. We can see that our RPF correctly reflects the structural roles. That is, the nodes $V_3, V_4, V_5$ in graph G and the nodes $V_1', V_3', V_5'$ in graph G' have the same structural role. And the nodes $V_1, V_2$ in graph G and the nodes $V_2', V_4'$ in graph G' have the same structural role.

implicit feature map. Then given a graph $G$, we can embed it into $\mathcal{H}$ in the following procedure,

$$G \rightarrow \mu_G \rightarrow m_G, \text{ and } m_G = \int_{\mathcal{A}} \phi \mathrm{d}\mu_G = \frac{1}{n} \sum_{i=1}^{n} \phi(\boldsymbol{p}_i, a_i^1, ..., a_i^L). \tag{7}$$

## 4.1 Graph kernels (I)

An important benefit of Hilbert space embedding of graphs is that it is straightforward to generalize the positive definite kernels defined on Euclidean spaces to the set of graphs.

Given two graphs $G$ and $H$, let $\{\triangle_i^G\}_{i=1}^{n_G}$ and $\{\triangle_i^H\}_{j=1}^{n_H}$ be the respective set representations $\left(\triangle_i^G = (\vec{\boldsymbol{p}}_i, a_i^1, a_i^2, ..., a_i^L)\right.$ and likewise $\left.\triangle_j^H\right)$. Let $\boldsymbol{K}_{GG}$, $\boldsymbol{K}_{HH}$, and $\boldsymbol{K}_{GH}$ be the kernel matrices, induced by the embedding kernel $k$. That is, they are defined such that $(\boldsymbol{K}_{GG})_{ij} = k(\triangle_i^G, \triangle_j^G)$, $(\boldsymbol{K}_{HH})_{ij} = k(\triangle_i^H, \triangle_j^H)$, and $(\boldsymbol{K}_{GH})_{ij} = k(\triangle_i^G, \triangle_j^H)$.

**Proposition 2.** *Let $\mathcal{G}$ be the set of graphs with attribute domains $\mathcal{A}_1, \mathcal{A}_2, ..., \mathcal{A}_L$. Let $G$ and $H$ be two graphs in $\mathcal{G}$. Let $m_G$ and $m_H$ be the corresponding graph embeddings. Then the following functions are positive definite graph kernels defined on $\mathcal{G} \times \mathcal{G}$.*

$$K_1(G, H) = (c + \langle m_G, m_H \rangle_{\mathcal{H}})^d = (c + \frac{1}{n_G n_H} \vec{\boldsymbol{1}}_{n_G}^T \boldsymbol{K}_{GH} \vec{\boldsymbol{1}}_{n_H})^d, c \geq 0, d \in \mathbb{N}, \tag{8a}$$

$$K_2(G, H) = \exp(-\gamma \|m_G - m_H\|_{\mathcal{H}}^p) = \exp\left[-\gamma \mathrm{MMD}^p(\mu_G, \mu_H)\right], \gamma > 0, 0 < p \leq 2, \tag{8b}$$

*where* $\mathrm{MMD}(\mu_G, \mu_H) = (\frac{1}{n_G^2} \vec{\boldsymbol{1}}_{n_G}^T \boldsymbol{K}_{GG} \vec{\boldsymbol{1}}_{n_G} + \frac{1}{n_H^2} \vec{\boldsymbol{1}}_{n_H}^T \boldsymbol{K}_{HH} \vec{\boldsymbol{1}}_{n_H} - \frac{2}{n_G n_H} \vec{\boldsymbol{1}}_{n_G}^T \boldsymbol{K}_{GH} \vec{\boldsymbol{1}}_{n_H})^{\frac{1}{2}}$ *is the maximum mean discrepancy (MMD) [11].*

**Kernel selection.** In real applications, such as bioinformatics, graphs may have discrete labels and (multi-dimensional) real-valued attributes. Hence, three attributes domains are involved in the computation of our graph kernels: the structural role attribute domain $\mathcal{A}_0$, the discrete attribute domain $\mathcal{A}_d$, and the continuous attribute domain $\mathcal{A}_c$. For $\mathcal{A}_d$, we can use the Delta kernel $k_d(a, b) = I_{\{a=b\}}$. For $\mathcal{A}_0$ and $\mathcal{A}_c$, which are just the Euclidean spaces, we can use the Gaussian RBF kernel, the Laplacian RBF kernel, or the polynomial kernel.

# 5 Approximated Hilbert space embedding of graphs

Based on the above discussions, we see that obtaining a graph kernel value between each pair of graphs requires calculating the inner product or the $L_2$ distance between two Hilbert embeddings (see (8a) and (8b)), both of which scale quadratically to the node numbers. Such time complexity precludes application to large graph datasets. To tackle the above issues, we employ the recently emerged approximate explicit feature maps [23].

For a kernel $k_l$ on the attribute domain $\mathcal{A}_l$, $l = 0, 1, ..., L$, we find an explicit map $\hat{\phi} : \mathcal{A}_l \rightarrow \mathbb{R}^{D_l}$, so that

$$\forall a, b \in \mathcal{A}_l, \langle \hat{\phi}(a), \hat{\phi}(b) \rangle = \hat{k}_l(a, b), \text{ and } \hat{k}_l(a, b) \rightarrow k_l(a, b) \text{ as } D_l \rightarrow \infty. \tag{9}$$

The explicit feature maps will be directly used to compute the approximate graph embeddings, by virtue of tensor algebra (see Section 2.2). The following theorem says that the approximate explicit graph embeddings can be written as the linear combination of rank-one tensors.

**Theorem 2.** *Let $G$ and $H$ be any two graphs in $\mathcal{G}$. Let $\{(\vec{\boldsymbol{p}}_i, a_i^1, a_i^2, ..., a_i^L)\}_{i=1}^{n_G}$ and $\{(\vec{\boldsymbol{q}}_j, b_j^1, b_j^2, ..., b_j^L)\}_{j=1}^{n_H}$ be the respective set representations of $G$ and $H$. Then their approximate explicit graph embeddings, $\hat{m}_G$ and $\hat{m}_H$, are tensors in $\mathbb{R}^{D_0 \times D_1 \times ... \times D_L}$, and can be written as*

$$\hat{m}_G = \frac{1}{n_G} \sum_{i=1}^{n_G} \hat{\phi}_0(\vec{\boldsymbol{p}}_i) \circ \hat{\phi}_1(a_i^1) \circ ... \circ \hat{\phi}_L(a_i^L), \quad \hat{m}_H = \frac{1}{n_H} \sum_{j=1}^{n_H} \hat{\phi}_0(\vec{\boldsymbol{q}}_j) \circ \hat{\phi}_1(b_j^1) \circ ... \circ \hat{\phi}_L(b_j^L). \tag{10}$$

*That is, as $D_0, D_1, ..., D_L \rightarrow \infty$, we have $\langle \hat{m}_G, \hat{m}_H \rangle_{\mathcal{T}} \rightarrow \langle m_G, m_H \rangle_{\mathcal{H}}$.*

## 5.1 Graph Kernels (II)

With approximate tensor embeddings (10), we obtain new graph kernels.

**Proposition 3.** *The following functions are positive definite graph kernels defined on $\mathcal{G} \times \mathcal{G}$.*

$$\hat{K}_1(G, H) = (c + \langle \hat{m}_G, \hat{m}_H \rangle_{\mathcal{T}})^d = \left[ c + \text{vec}(\hat{m}_{\mathcal{G}})^T \text{vec}(\hat{m}_{\mathcal{H}}) \right]^d, c \geq 0, d \in \mathbb{N}, \quad (11a)$$

$$\hat{K}_2(G, H) = \exp(-\gamma \|\hat{m}_G - \hat{m}_H\|_{\mathcal{T}}^p) = \exp(-\gamma \|\text{vec}(\hat{m}_G) - \text{vec}(\hat{m}_H)\|_2^p), \gamma > 0, 0 < p \leq 2.. \quad (11b)$$

*Moreover, as $D_0, D_1, ..., D_L \to \infty$, we have $\hat{K}_1(G, H) \to K_1(G, H)$ and $\hat{K}_2(G, H) \to K_2(G, H)$.*

The vectorization of $\hat{m}_G$ (or $\hat{m}_H$) can be easily implemented by the Kronecker product, i.e., $\text{vec}(\hat{m}_G) = \frac{1}{n_G} \sum_{i=1}^{n_G} \hat{\phi}_0(\vec{p}_i) \otimes \hat{\phi}_1(a_i^1) \otimes ... \otimes \hat{\phi}_L(a_i^L)$. To obtain above graph kernels, we need only to compute the Euclidean inner product or distance between vectors. More notably, the size of the tensor representation does not depends on node numbers, making it scalable to large graphs.

**Approximate explicit feature map selection.** For the Delta kernel on the discrete attribute domain, we directly use the one-hot vector. For shift-invariant kernels, i.e., $k(\vec{x}, \vec{y}) = k(\vec{x} - \vec{y})$, on Euclidean spaces, e.g., $\mathcal{A}_0$ and $\mathcal{A}_c$, we make use of random Fourier feature map [23], $\hat{\phi} : \mathbb{R}^d \to \mathbb{R}^D$, satisfying $\langle \hat{\phi}(\vec{x}), \hat{\phi}(\vec{y}) \rangle \approx k(\vec{x}, \vec{y})$. To do this, we first draw $D$ i.i.d. samples $\omega_1, \omega_2, ..., \omega_D$ from a proper distribution $p(\omega)$. (Note that in this paper, we use $p(\omega) = \frac{1}{(\sqrt{2\pi}\sigma)^D} \exp(-\frac{\|\omega\|_2^2}{2\sigma^2})$.) Next, we draw $D$ i.i.d. samples $b_1, b_2, ..., b_D$ from the uniform distribution on $[0, 2\pi]$. Finally, we can calculate $\hat{\phi}(\vec{x}) = \sqrt{\frac{2}{D}} \left[ \cos(\omega_1^T \vec{x} + b_1), ..., \cos(\omega_D^T \vec{x} + b_D) \right]^T \in \mathbb{R}^D$.

# 6 Experiments

In this section, we conduct extensive experiments to demonstrate the effectiveness of our graph kernels. We run all the experiments on a laptop with an Intel i7-7820HQ, 2.90GHz CPU and 64GB RAM. We implement our algorithms in Matlab, except for the Monte Carlo based computation of RPF (see Section 3.2,1), which is implemented in C++.

## 6.1 Datasets

We conduct graph classification on four types of benchmark datasets [16]. (i) Non-attributed (unlabeled) graphs datasets: COLLAB, IMDB-BINARY, IMDB-MULTI, REDDIT-BINARY, REDDIT-MULTI(5K), and REDDIT-MULTI(12K) [31] are generated from social networks. (ii) Graphs with discrete attributes (labels): DD [8] are proteins. MUTAG [7], NCI1 [25], PTC-FM, PTC-FR, PTC-MM, and PTC-MR [13] are chemical compounds. (iii) Graphs with continuous attributes: FRANK is a chemical molecule dataset [15]. SYNTHETIC and Synthie are synthetic datasets based on random graphs, which were first introduced in [9] and [19], respectively. (iv) Graphs with both discrete and continuous attributes: ENZYMES and PROTEINS [2] are graph representations of proteins. BZR, COX2, and DHFR [27] are chemical compounds. Detailed descriptions, including statistical properties, of these 21 datasets are provided in the supplementary material.

## 6.2 Experimental setup

We demonstrate both the graph kernels (I) and (II) introduced in Section 4.1 and Section 5.1, which are denoted by $\text{RetGK}_{\text{I}}$ and $\text{RetGK}_{\text{II}}$, respectively. The Monte Carlo computation of return probability features, denoted by $\text{RetGK}_{\text{II}}(\text{MC})$, is also considered. In our experiments, we repeat 200 Monte Carlo trials, i.e., $M = 200$, for obtaining RPF. For handling the isolated nodes, whose degrees are zero, we artificially add a self-loop for each node in graphs.

**Parameters.** In all experiments, we set the random walk step $S = 50$. For $\text{RetGK}_{\text{I}}$, we use the Laplacian RBF kernel for both the structural role domain $\mathcal{A}_0$, and the continuous attribute domain $\mathcal{A}_c$, i.e., $k_0(\vec{p}, \vec{q}) = \exp(-\gamma_0 \|\vec{p} - \vec{q}\|_2)$ and $k_c(\vec{a}, \vec{b}) = \exp(-\gamma_c \|\vec{a} - \vec{b}\|_2)$. We set $\gamma_0$ to be the inverse of the median of all pairwise distances, and set $\gamma_c$ to be the inverse of the square root of the attributes' dimension, except for the FRANK dataset, whose $\gamma_c$ is set to be the recommended value $\sqrt{0.0073}$

in the paper [22] and [19]. For $\mathrm{RetGK_{II}}$, on the first three types of graphs, we set the dimensions of random Fourier feature maps on $\mathcal{A}_0$ and $\mathcal{A}_c$ both to be 200, i.e., $D_0 = D_c = 200$, except for the FRANK dataset, whose $D_c$ is set to be 500 because its attributes lie in a much higher dimensional space. On the graphs with both discrete and continuous attributes, for the sake of computational efficiency, we set $D_0 = D_c = 100$. For both $\mathrm{RetGK_I}$ and $\mathrm{RetGK_{II}}$, we make use of the graph kernels with exponential forms, $\exp(-\gamma \| \cdot \|^p)$, (see (8b) and (11b)). We select $p$ from $\{1, 2\}$, and set $\gamma = \frac{1}{\mathrm{dist}^p}$, where $\mathrm{dist}$ is the median of all the pairwise graph embedding distances.

We compare our graph kernels with many state-of-the-art graph classification algorithms: (i) the shortest path kernel (SP) [1], (ii) the Weisfeiler-Lehman subtree kernel (WL) [25], (iii) the graphlet count kernel (GK)[26], (iv) deep graph kernels (DGK) [31], (v) PATCHY-SAN convolutional neural network (PSCN) [20], (vi) deep graph convolutional neural network (DGCNN) [33], (vii) graph invariant kernels (GIK) [22], (viii) hashing Weisfeiler-Lehman graph kernels (HGK(WL)) [19], and (IX) subgraph matching kernels (CSM) [18].

For all kinds of graph kernels, we employ SVM [3] as the final classifier. The tradeoff parameter $C$ is selected from $\{10^{-3}, 10^{-2}, 10^{-1}, 1, 10, 10^2, 10^3\}$. We perform 10-fold cross-validations, using 9 folds for training and 1 for testing, and repeat the experiments 10 times. We report average classification accuracies and standard errors.

## 6.3 Experimental Results

The classification results[4] on four types of datasets are shown in Tables 1, 2, 3, and 4. The best results are highlighted in bold. We also report the total time of computing the graph kernels of all the datasets in each table. It can be seen that graph kernels $\mathrm{RetGK_I}$ and $\mathrm{RetGK_{II}}$ both achieve superior or comparable performance on all the benchmark datasets. Especially on the datasets COLLAB, REDDIT-BINARY, REDDIT-MULTI(12K), Synthie, BZR, COX2, our approaches significantly outperform other state-of-the-art algorithms. The classification accuracies of our approaches on these datasets are at least six percentage points higher than those of the best baseline algorithms. Moreover, we see that $\mathrm{RetGK_{II}}$ and $\mathrm{RetGK_{II}}$(MC) are faster than baseline methods. Their running times remain perfectly practical. On the large social network datasets (see Table 1), $\mathrm{RetGK_{II}}$(MC) is almost one order of magnitude faster than the Weisfeiler-Lehman subtree kernel, which is well known for its computational efficiency.

## 6.4 Sensitivity analysis

Here, we conduct a parameter sensitivity analysis of $\mathrm{RetGK_{II}}$ on the datasets REDDIT-BINARY, NCI1, SYNTHETIC, Synthie, ENZYMES, and PROTEINS. We test the stability of $\mathrm{RetGK_{II}}$ by varying the values of the random walk steps $S$, the dimension $D_0$ of the approximate explicit feature map on $\mathcal{A}_0$, and the dimension $D_c$ of the feature map on $\mathcal{A}_c$. We plot the average classification accuracy of ten repetitions of 10-fold cross-validations with respect to $S$, $D_0$, and $D_c$ in Fig. 3. It can be concluded that $\mathrm{RetGK_{II}}$ performs consistently across a wide range of parameter values.

Table 1: Classification results (in %) for non-attributed (unlabeled) graph datasets

| Datasets | WL | GK | DGK | PSCN | $\mathrm{RetGK_I}$ | $\mathrm{RetGK_{II}}$ | $\mathrm{RetGK_{II}}$(MC) |
|---|---|---|---|---|---|---|---|
| COLLAB | 74.8(0.2) | 72.8(0.3) | 73.1(0.3) | 72.6(2.2) | **81.0(0.3)** | 80.6(0.3) | 73.6(0.3) |
| IMDB-BINARY | 70.8(0.5) | 65.9(1.0) | 67.0(0.6) | 71.0(2.3) | 71.9(1.0) | **72.3(0.6)** | 71.0(0.6) |
| IMDB-MULTI | **49.8(0.5)** | 43.9(0.4) | 44.6(0.5) | 45.2(2.8) | 47.7(0.3) | 48.7(0.6) | 46.7(0.6) |
| REDDIT-BINARY | 68.2(0.2) | 77.3(0.2) | 78.0(0.4) | 86.3(1.6) | **92.6(0.3)** | 91.6(0.2) | 90.8(0.2) |
| REDDIT-MULTI(5K) | 51.2(0.3) | 41.0(0.2) | 41.3(0.2) | 49.1(0.7) | **56.1(0.5)** | 55.3(0.3) | 54.2(0.3) |
| REDDIT-MULTI(12K) | 32.6(0.3) | 31.8(0.1) | 32.2(0.1) | 41.3(0.4) | **48.7(0.2)** | 47.1(0.3) | 45.9(0.2) |
| Total time | 2h3m | – | – | – | 48h14m | 17m14s | **6m9s** |

Table 2: Classification results (in %) for graph datasets with discrete attributes

| Datasets | SP | WL | GK | CSM | DGCNN | DGK | PSCN | RetGK$_\mathrm{I}$ | RetGK$_\mathrm{II}$ |
|---|---|---|---|---|---|---|---|---|---|
| ENZYMES | 38.6(1.5) | 53.4(0.9) | – | **60.4(1.6)** | – | 53.4(0.9) | – | **60.4(0.8)** | 59.1(1.1) |
| PROTEINS | 73.3(0.9) | 71.2(0.8) | 71.7(0.6) | – | 75.5(0.9) | 75.7(0.5) | 75.0(2.5) | **75.8(0.6)** | 75.2(0.3) |
| MUTAG | 85.2(2.3) | 84.4(1.5) | 81.6(2.1) | 85.4(1.2) | 85.8(1.7) | 87.4(2.7) | 89.0(4.4) | **90.3(1.1)** | 90.1(1.0) |
| DD | >24h | 78.6(0.4) | 78.5(0.3) | – | 79.4(0.9) | – | 76.2(2.6) | **81.6(0.3)** | 81.0(0.5) |
| NCI1 | 74.8(0.4) | **85.4(0.3)** | 62.3(0.3) | – | 74.4(0.5) | 80.3(0.5) | 76.3(1.7) | 84.5(0.2) | 83.5(0.2) |
| PTC-FM | 60.5(1.7) | 55.2(2.3) | – | 63.8(1.0) | – | – | – | 62.3(1.0) | **63.9(1.3)** |
| PTC-FR | 61.6(1.0) | 63.9(1.4) | – | 65.5(1.4) | – | – | – | 66.7(1.4) | **67.8(1.1)** |
| PTC-MM | 62.9(1.4) | 60.6(1.1) | – | 63.3(1.7) | – | – | – | 65.6(1.1) | **67.9(1.4)** |
| PTC-MR | 57.8(2.1) | 55.4(1.5) | 57.3(1.1) | 58.1(1.6) | 58.6(2.5) | 60.1(2.6) | 62.3(5.7) | **62.5(1.6)** | 62.1(1.5) |
| Total time | >24h | 2m27s | – | – | – | – | – | 38m4s | **49.9s** |

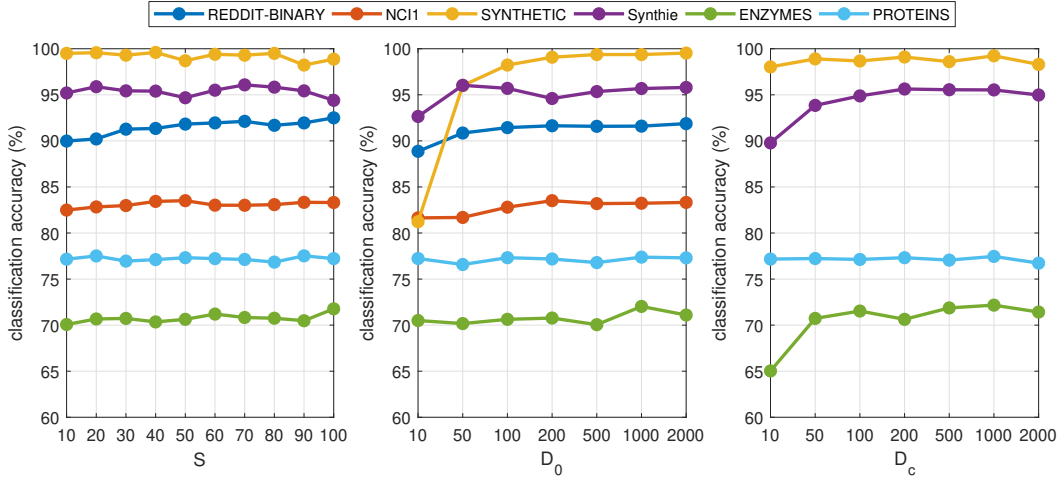

Figure 3: Parameter sensitivity study for RetGK$_\mathrm{II}$ on six benchmark datasets

Table 3: Classification results (in %) for graph datasets with continuous attributes

| Datasets | HGK(WL) | RetGK$_\mathrm{I}$ | RetGK$_\mathrm{II}$ |
|---|---|---|---|
| ENZYMES | 63.9(1.1) | 70.0(0.9) | **70.7(0.9)** |
| PROTEINS | 74.9(0.6) | **76.2(0.5)** | 75.9(0.4) |
| FRANK | 73.2(0.3) | 76.4(0.3) | **76.7(0.4)** |
| SYNTHETIC | 97.6(0.4) | 97.9(0.3) | **98.9(0.4)** |
| Synthie | 80.3(1.4) | **97.1(0.3)** | 96.2(0.3) |
| Total time | – | 45m30s | **40.8s** |

Table 4: Classification results (in %) for graph datasets with both discrete and continuous attributes

| Datasets | GIK | CSM | RetGK$_\mathrm{I}$ | RetGK$_\mathrm{II}$ |
|---|---|---|---|---|
| ENZYMES | 71.7(0.8) | 69.8(0.7) | **72.2(0.8)** | 70.6(0.7) |
| PROTEINS | 76.1(0.3) | – | **78.0(0.3)** | 77.3(0.5) |
| BZR | – | 79.4(1.2) | 86.4(1.2) | **87.1(0.7)** |
| COX2 | – | 74.4(1.7) | 80.1(0.9) | **81.4(0.6)** |
| DHFR | – | 79.9(1.1) | 81.5(0.9) | **82.5(0.8)** |
| Total time | – | – | 4m17s | **2m51s** |

## 7 Conclusion

In this paper, we introduced the return probability feature for characterizing and comparing the structural role of nodes across graphs. Based on the RPF, we embedded graphs in an RKHS and derived the corresponding graph kernels RetGK$_\mathrm{I}$. Then, making use of approximate explicit feature maps, we represented each graph with a multi-dimensional tensor, and then obtained the computationally efficient graph kernels RetGK$_\mathrm{II}$. We applied RetGK$_\mathrm{I}$ and RetGK$_\mathrm{II}$ to classify graphs, and achieved promising results on many benchmark datasets. Given the prevalence of structured data, we believe that our work can be potentially useful in many applications.

## 8 Acknowledgement

This work was supported in part by the AFOSR grant FA9550-16-1-0386.

## Footnotes

[1]In the literature, the discrete node attributes are usually called "labels".

[2]A vector $\vec{\boldsymbol{u}} \in \mathbb{R}^D$ is a first-order tensor, and a matrix $\boldsymbol{A} \in \mathbb{R}^{D_1 \times D_2}$ is a second-order tensor.

[3]Note that since the signs of these eigenvectors are not fixed, we use the absolute value as in [21]

[4]The accuracies of WL, SP and GK are obtained from our own experiments. For others competing algorithms, we directly quote the values from their papers.

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
