[Supplementary Material]

# RetGK: Graph Kernels based on Return Probabilities of Random Walks: Supplementary Material

**Zhen Zhang, Mianzhi Wang, Yijian Xiang, Yan Huang, and Arye Nehorai**
Department of Electrical and Systems Engineering
Washington University in St. Louis
St. Louis, MO 63130
{zhen.zhang, mianzhi.wang, yijian.xiang, yanhuang640, nehorai}@wustl.edu

## Abstract

The supplementary material consists of two parts. In the first part, we prove all the mathematical results presented in the paper. In the second part, we provide the detailed descriptions of the benchmark graph datasets used in the paper.

## 1 Proofs

### 1.1 Proving Proposition 1

**Proposition 1.** *Let $G$ and $H$ be two isomorphic graphs of $n$ nodes, and let $\tau : \{1, 2, ..., n\} \to \{1, 2, ..., n\}$ be the corresponding isomorphism. Then,*

$$\forall v_i \in V_G, \ s = 1, 2, ..., \infty, \ \boldsymbol{P}_G^s(i, i) = \boldsymbol{P}_H^s(\tau(i), \tau(i)). \tag{1}$$

*Proof.* Let $\boldsymbol{\Pi}$ be the permutation matrix induced by $\tau$, i.e., $\boldsymbol{\Pi}(i, j) = \delta_{j=\tau(i)}$, then, $\boldsymbol{\Pi}^T\boldsymbol{\Pi} = \boldsymbol{\Pi}\boldsymbol{\Pi}^T = \boldsymbol{I}_n$. Since $G$ and $H$ are isomorphic, we have $\boldsymbol{A}_H(\tau(i), \tau(j)) = \boldsymbol{A}_G(i, j)$, and $\boldsymbol{D}_H(\tau(i), \tau(i)) = \boldsymbol{D}_G(i, i)$, which is equivalent with $\boldsymbol{A}_H = \boldsymbol{\Pi}^T\boldsymbol{A}_G\boldsymbol{\Pi}$ and $\boldsymbol{D}_H = \boldsymbol{\Pi}^T\boldsymbol{D}_G\boldsymbol{\Pi}$.

Then $\boldsymbol{P}_H = \boldsymbol{D}_H^{-1}\boldsymbol{A}_H = (\boldsymbol{\Pi}^T\boldsymbol{D}_G\boldsymbol{\Pi})^{-1}(\boldsymbol{\Pi}^T\boldsymbol{A}_G\boldsymbol{\Pi}) = \boldsymbol{\Pi}^T\boldsymbol{P}_G\boldsymbol{\Pi}$. So $\boldsymbol{P}_H^s = (\boldsymbol{\Pi}^T\boldsymbol{P}_G\boldsymbol{\Pi})^s = \boldsymbol{\Pi}^T\boldsymbol{P}_G^s\boldsymbol{\Pi}$, which implies $\boldsymbol{P}_G^s(i, i) = \boldsymbol{P}_H^s(\tau(i), \tau(i))$. $\square$

### 1.2 Proving Theorem 1

**Theorem 1.** *Let $G$ and $H$ be two connected graphs of the same size $n$ and volume* Vol, *and let $\boldsymbol{P}_G$ and $\boldsymbol{P}_H$ be the corresponding transition probability matrices. Let $\{(\lambda_k, \vec{\psi}_k)\}_{k=1}^n$ and $\{(\mu_k, \vec{\varphi}_k)\}_{k=1}^n$ be eigenpairs of $\boldsymbol{P}_G$ and $\boldsymbol{P}_H$, respectively. Let $\tau : \{1, 2, ..., n\} \to \{1, 2, ..., n\}$ be a permutation map. If $\boldsymbol{P}_G^s(i, i) = \boldsymbol{P}_H^s(\tau(i), \tau(i)), \forall v_i \in V_G, \forall s = 1, 2, ..., n, i.e.,$ $\mathrm{RPF}_G^n = \mathrm{RPF}_H^n$, then,*

1. $\mathrm{RPF}_G^S = \mathrm{RPF}_H^S, \forall S = n + 1, n + 2, ..., \infty$;
2. $\{\lambda_1, \lambda_2, ..., \lambda_n\} = \{\mu_1, \mu_2, ..., \mu_n\}$;
3. *If the eigenvalues sorted by their magnitudes satisfy: $|\lambda_1| > |\lambda_2| > ... > |\lambda_m| > 0$, $|\lambda_{m+1}| = ... = |\lambda_n| = 0$, then we have that $|\vec{\psi}_k(i)| = |\vec{\varphi}_k(\tau(i))|, \forall v_i \in V_G, \forall k = 1, 2, ..., m$.*

*We first present some useful lemmas.*

**Lemma 1.** *(Cayley-Hamilton Theorem, see[11, 16]) Let $A$ be an $n \times n$ matrix, and let $P(\lambda) = \det(\lambda I_n - A)$ be the corresponding characteristic polynomial of $A$, then $P(A) = 0$, i.e.,*

$$A^n + c_{n-1}A^{n-1} + \cdots + c_1 A + (-1)^n \det(A) I_n = 0, \tag{2}$$

*where $c_{n-k} = \frac{(-1)^k}{k!} B_k\big(s_1, (-1)s_2, 2!s_3, \cdots, (-1)^{k-1}(k-1)!s_k\big)$, and $B_k$ is the Bell polynomial and $s_i = \text{trace}(A^i)$. In particular, $\det(A) = \frac{1}{n!} B_n\big(s_1, (-1)s_2, 2!s_3, \cdots, (-1)^{n-1}(n-1)!s_n\big)$.*

**Remark 1.** *We observe that all the coefficients in (2) are determined by $\text{trace}(A)$, $\text{trace}(A^2), \cdots, \text{trace}(A^n)$.*

**Corollary 1.** *Let $A$ and $B$ be two $n \times n$ matrices. If $\text{trace}(A^k) = \text{trace}(B^k)$, $k = 1, 2, \cdots, n$, then $A$ and $B$ have the same eigenvalues set.*

*Proof.* Let $P^A(\lambda)$ and $P^B(\lambda)$ be the characteristic polynomials of $A$ and $B$ respectively, and let $c_{n-k}^A$ and $c_{n-k}^B$, $k = 1, 2, \cdots, n-1$ be the corresponding coefficients. Since $\text{trace}(A^k) = \text{trace}(B^k)$, $k = 1, 2, \cdots, n$, we have that $c_{n-k}^A = c_{n-k}^B$, $k = 1, 2, \cdots, n-1$, and $\det(A) = \det(B)$. Therefore, the roots of $P^A(\lambda)$ and $P^B(\lambda)$ are the same, which is equivalent to $A$ and $B$ having the same eigenvalues set. $\square$

**Corollary 2.** *Let $A$ and $B$ be two $n \times n$ matrices. If $A^s(i,i) = B^s(i,i)$, $s = 1, 2, \cdots, n$, $i = 1, 2, \cdots, n$, then $A^s(i,i) = B^s(i,i)$, $s = n+1, n+2, \cdots$, $i = 1, 2, \cdots, n$.*

*Proof.* It is easy to obtain $\text{trace}(A^s) = \text{trace}(B^s)$, $s = 1, 2, \cdots, n$. Then based on the lemma 1, the characteristic polynomials of $A$ and $B$ are same. Moreover,

$$A^n = -c_{n-1}A^{n-1} - c_{n-2}A^{n-2} - \cdots - c_1 A - (-1)^n \det(A) I_n. \tag{3}$$

Multiply $A^{s-n}$, $s \geq n+1$ on both sides, and we have

$$A^s = -c_{n-1}A^{s-1} - c_{n-2}A^{s-2} - \cdots - c_1 A^{s-(n-1)} - (-1)^n \det(A) A^{s-n}. \tag{4}$$

Immediately, for any $i = 1, 2, \cdots, n$,

$$A^s(i,i) = -c_{n-1}A^{s-1}(i,i) - c_{n-2}A^{s-2}(i,i) - \cdots - c_1 A^{s-(n-1)}(i,i) - (-1)^n \det(A) A^{s-n}(i,i). \tag{5}$$

From the iterative formula (5), we can see that $A^s(i,i)$, $s = n+1, n+2, \cdots$ are uniquely determined by $\{A(i,i), A^2(i,i), \cdots, A^n(i,i)\}$. Similarly, $B^s(i,i)$, $s = n+1, n+2, \cdots$ are uniquely determined by $\{B(i,i), B^2(i,i), \cdots, B^n(i,i)\}$. Combining with the fact $A^s(i,i) = B^s(i,i)$, $s = 1, 2, .., n$, we obtain the desired result. $\square$

**Lemma 2.** *(Time-reversible Markov chains, see [2]) If for an irreducible Markov chain with transition matrix $P$, there exists a probability solution $\vec{\pi}$ to the "Time-reversibility" set of equations,*

$$\vec{\pi}_i^T P(i,j) = \vec{\pi}_j^T P(j,i), \tag{6}$$

*for all pairs of states $i, j$, then the solution $\vec{\pi}$ is the unique stationary distribution, i.e., $\lim_{s \to +\infty} P^s(i,j) = \vec{\pi}_j$.*

**Remark 2.** *For a connected graph $G$, the random walk defined on it can be considered as a irreducible Markov chain. We define a probability vector $\vec{\pi}$ as $\vec{\pi}_i = \frac{D_G(i,i)}{\text{Vol}_G}$, where $\text{Vol}_G$ is the volume of the graph, i.e., $\text{Vol}_G = \sum_{i=1}^n D_G(i,i)$. Then we have,*

$$\vec{\pi}_i^T P_G(i,j) = \frac{D_G(i,i)}{\text{Vol}_G} \times \frac{w_{ij}}{D_G(i,i)} = \frac{D_G(j,j)}{\text{Vol}_G} \times \frac{w_{ji}}{D_G(j,j)} = \vec{\pi}_j^T P_G(j,i). \tag{7}$$

*Therefore, $\vec{\pi}$ defined above is the stationary distribution of the random walk.*

***Now we prove theorem 1.***

*Proof.* **(1).** Let $\Pi$ be the permutation matrix induced by $\tau$, i.e., $\Pi(i,j) = \delta_{j=\tau(i)}$. Then we have $\forall i, j = 1, 2, \cdots, n$, $P_H(\tau(i), \tau(j)) = (\Pi P_H \Pi^T)(i,j)$. Since $P_G^s(i,i) = P_H^s(\tau(i), \tau(i)) = (\Pi P_H^s \Pi^T)(i,i) = (\Pi P_H \Pi^T)^s(i,i)$, $\forall v_i \in V_G$, and $\forall s = 1, 2, \cdots, n$, by Corollary 2, we have

$P_G^s(i,i) = (\mathbf{\Pi} P_H \mathbf{\Pi}^T)^s(i,i) = P_H^s(\tau(i), \tau(i))$, $\forall s = n+1, n+2 \cdots + \infty$. Now, the first conclusion has been proved.

$(\mathbf{2})$. The second one can be directly concluded from corollary 1.

$(\mathbf{3})$. Let $\boldsymbol{D}_G$ and $\boldsymbol{D}_H$ be the degree matrices of graph $G$ and $H$, respectively. Then by Remark 2,

$$\frac{\boldsymbol{D}_G(i,i)}{\text{Vol}} = \lim_{s \to +\infty} \boldsymbol{P}_G^s(i,i) = \lim_{s \to +\infty} \boldsymbol{P}_H^s(\tau(i), \tau(i)) = \frac{\boldsymbol{D}_H(\tau(i), \tau(i))}{\text{Vol}}. \tag{8}$$

So

$$\boldsymbol{D}_G(i,i) = \boldsymbol{D}_H(\tau(i), \tau(i)), \forall v_i \in V_G. \tag{9}$$

Let $\boldsymbol{A}_G$ and $\boldsymbol{A}_H$ be the adjacent matrices of $G$ and $H$ respectively, and write $\boldsymbol{P}_G = \boldsymbol{D}_G^{-1}\boldsymbol{A}_G = \boldsymbol{D}_G^{-\frac{1}{2}}(\boldsymbol{D}_G^{-\frac{1}{2}}\boldsymbol{A}_G\boldsymbol{D}_G^{-\frac{1}{2}})\boldsymbol{D}_G^{\frac{1}{2}}$. Let $\boldsymbol{B}_G = \boldsymbol{D}_G^{-\frac{1}{2}}\boldsymbol{A}_G\boldsymbol{D}_G^{-\frac{1}{2}} \implies \boldsymbol{P}_G = \boldsymbol{D}_G^{-\frac{1}{2}}\boldsymbol{B}_G\boldsymbol{D}_G^{\frac{1}{2}} \implies \boldsymbol{P}_G^s = \boldsymbol{D}_G^{-\frac{1}{2}}\boldsymbol{B}_G^s\boldsymbol{D}_G^{\frac{1}{2}} \implies \boldsymbol{P}_G^s(i,i) = \boldsymbol{B}_G^s(i,i)$. $\boldsymbol{B}_G$ is a symmetric matrix, and has the same eigenvalues as $\boldsymbol{P}_G$. Write the orthonormal eigen-decomposition of $\boldsymbol{B}_G$ as $\boldsymbol{B}_G = \sum_{k=1}^n \lambda_k \vec{u}_k \vec{u}_k^T$, then

$$\boldsymbol{P}_G^s(i,i) = \boldsymbol{B}_G^s(i,i) = \sum_{k=1}^n \lambda_k^s \vec{u}_k(i)^2, \tag{10}$$

where $\vec{u}_k(i)$ denotes the $i$th component of the eigenvector $\vec{u}_k$. Similarly, we have

$$\boldsymbol{P}_H^s(\tau(i), \tau(i)) = \boldsymbol{B}_H^s(\tau(i), \tau(i)) = \sum_{k=1}^n \mu_k^s \vec{v}_k(\tau(i))^2 = \sum_{k=1}^n \lambda_k^s \vec{v}_k(\tau(i))^2, \tag{11}$$

where $\vec{v}_k(\tau(i))$ denotes the $\tau(i)$th component of $\vec{v}_k$, and $\vec{v}_k$ is the $k$th eigenvector of $\boldsymbol{B}_H = \boldsymbol{D}_H^{-\frac{1}{2}}\boldsymbol{A}_H\boldsymbol{D}_H^{-\frac{1}{2}}$. The last equality of (11) holds because $\{\lambda_1, \lambda_2, ..., \lambda_n\} = \{\mu_1, \mu_2, ..., \mu_n\}$.

Next, we use mathematical induction to show that $|\vec{u}_k(i)| = |\vec{v}_k(\tau(i))|$, $\forall v_i \in V_G$, $\forall k = 1, 2, \cdots, m$.

**Step1:** For $k = 1$, $\boldsymbol{P}_G\vec{\mathbf{1}} = \vec{\mathbf{1}} \Leftrightarrow \boldsymbol{B}_G\boldsymbol{D}_G^{\frac{1}{2}}\vec{\mathbf{1}} = \boldsymbol{D}_G^{\frac{1}{2}}\vec{\mathbf{1}} \Leftrightarrow \vec{u}_1 = \pm\boldsymbol{D}_G^{\frac{1}{2}}\vec{\mathbf{1}}/\sqrt{\text{Vol}}$. Similarly, we have $\vec{v}_1 = \pm\boldsymbol{D}_H^{\frac{1}{2}}\vec{\mathbf{1}}/\sqrt{\text{Vol}}$. Since $\forall v_i \in V_G$, $\boldsymbol{D}_G(i,i) = \boldsymbol{D}_H(\tau(i), \tau(i))$, we have $|\vec{u}_1(i)| = |\vec{v}_1(\tau(i))|$, $\forall v_i \in V_G$.

**Step2:** We show that if the first $k$ eigenvectors satisfy, $|\vec{u}_1(i)| = |\vec{v}_1(\tau(i))|$, $|\vec{u}_2(i)| = |\vec{v}_2(\tau(i))|, \cdots$, $|\vec{u}_k(i)| = |\vec{v}_k(\tau(i))|$, then $|\vec{u}_{k+1}(i)| = |\vec{v}_{k+1}(\tau(i))|$, $\forall v_i \in V_G$, $\forall k = 1, 2, ..., m-2$.

We suppose that $\exists v_{i^*} \in V_G$, such that $|\vec{u}_{k+1}(i^*)| \neq |\vec{v}_{k+1}(\tau(i^*))|$. Without loss of generality, we assume that $\vec{u}_{k+1}(i^*)^2 - \vec{v}_{k+1}(\tau(i^*))^2 = \epsilon > 0$. Write

$$\begin{aligned}
&\boldsymbol{P}_G^s(i^*, i^*) - \boldsymbol{P}_H^s(\tau(i^*), \tau(i^*)) \\
&= \sum_{l=1}^n \lambda_l^s \vec{u}_l(i^*)^2 - \sum_{l=1}^n \lambda_l^s \vec{v}_l(\tau(i^*))^2 \\
&= \sum_{l=1}^m \lambda_l^s \vec{u}_l(i^*)^2 - \sum_{l=1}^m \lambda_l^s \vec{v}_l(\tau(i^*))^2 \\
&= \sum_{l=k+1}^m \lambda_l^s \vec{u}_l(i^*)^2 - \sum_{l=k+1}^m \lambda_l^s \vec{v}_l(\tau(i^*))^2 \\
&= \lambda_{k+1}^s \left[\vec{u}_{k+1}(i^*)^2 - \vec{v}_{k+1}(\tau(i^*))^2\right] - \sum_{l=k+2}^m \lambda_l^s \left[\vec{u}_l(i^*)^2 - \vec{v}_l(\tau(i^*))^2\right] \\
&= \lambda_{k+1}^s \left(\epsilon - \sum_{l=k+2}^m (\frac{\lambda_l}{\lambda_{k+1}})^s \left[\vec{u}_l(i^*)^2 - \vec{v}_l(\tau(i^*))^2\right]\right),
\end{aligned} \tag{12}$$

where the second equality holds because $|\lambda_1| > |\lambda_2| > ... > |\lambda_m| > 0$, $|\lambda_{m+1}| = ... = |\lambda_n| = 0$. With the fact that $|\vec{u}_l(i^*)^2 - \vec{v}_l(\tau(i^*))^2| \leq 1$ $\left(\text{since } 0 \leq \vec{u}_l(i^*)^2, \vec{v}_l(\tau(i^*))^2 \leq 1\right)$, and $|\frac{\lambda_l}{\lambda_{k+1}}| < 1$,

we have that there is a positive integer, $M$, such that,

$$\epsilon - \sum_{l=k+2}^{m} (\frac{\lambda_l}{\lambda_{k+1}})^{2M} \big[\vec{\boldsymbol{u}}_l(i^*)^2 - \vec{\boldsymbol{v}}_l(\tau(i^*))^2\big] > 0. \tag{13}$$

Therefore, $\boldsymbol{P}_G^{2M}(i^*, i^*) - \boldsymbol{P}_H^{2M}(\tau(i^*), \tau(i^*)) > 0$, contradicting the fact that $\boldsymbol{P}_G^s(i, i) = \boldsymbol{P}_H^s(\tau(i), \tau(i)), \forall v_i \in V_G, \forall s = 1, 2, ..., \infty$. So $|\vec{\boldsymbol{u}}_{k+1}(i)| = |\vec{\boldsymbol{v}}_{k+1}(\tau(i))|, \forall v_i \in V_G$.

**Step 3:** We show that if $|\vec{\boldsymbol{u}}_1(i)| = |\vec{\boldsymbol{v}}_1(\tau(i))|, |\vec{\boldsymbol{u}}_2(i)| = |\vec{\boldsymbol{v}}_2(\tau(i))|, \cdots, |\vec{\boldsymbol{u}}_{m-1}(i)| = |\vec{\boldsymbol{v}}_{m-1}(\tau(i))|$, then $|\vec{\boldsymbol{u}}_m(i)| = |\vec{\boldsymbol{v}}_m(\tau(i))|, \forall v_i \in V_G$. Since

$$0 = \boldsymbol{P}_G^s(i^*, i^*) - \boldsymbol{P}_H^s(\tau(i^*), \tau(i^*)) = \lambda_m^s \big[\vec{\boldsymbol{u}}_m(i^*)^2 - \vec{\boldsymbol{v}}_m(\tau(i^*))^2\big], \tag{14}$$

and $\lambda_m^s \neq 0$, we immediately have that $|\vec{\boldsymbol{u}}_m(i^*)| = |\vec{\boldsymbol{v}}_m(\tau(i^*))|$.

Combining all these three steps, we obtain the desired result $|\vec{\boldsymbol{u}}_k(i)| = |\vec{\boldsymbol{v}}_k(\tau(i))|$, $\forall v_i \in V_G$, $\forall k = 1, 2, \cdots, m$.

Since $\boldsymbol{P}_G = \boldsymbol{D}_G^{-\frac{1}{2}} \boldsymbol{B}_G \boldsymbol{D}_G^{\frac{1}{2}}$, we have the fact that $(\lambda_k, \vec{\boldsymbol{u}}_k)$ is an eigenpair of $\boldsymbol{B}_G$ if and only if $(\lambda_k, \boldsymbol{D}_G^{-\frac{1}{2}}\vec{\boldsymbol{u}}_k)$ is an eigenpair of $\boldsymbol{P}_G$. The above implies that $\vec{\boldsymbol{\psi}}_k = \boldsymbol{D}_G^{-1}\vec{\boldsymbol{u}}_k$, and similarly $\vec{\boldsymbol{\varphi}}_k = \boldsymbol{D}_H^{-1}\vec{\boldsymbol{v}}_k$. Now, $\forall v_i \in V_G, \forall k = 1, 2, ..., m$, we have

$$|\vec{\boldsymbol{\varphi}}_k(\tau(i))| = \boldsymbol{D}_H^{-1}(\tau(i))|\vec{\boldsymbol{v}}_k(\tau(i))| = \boldsymbol{D}_G^{-1}(i)|\vec{\boldsymbol{v}}_k(\tau(i))| = \boldsymbol{D}_G^{-1}(i)|\vec{\boldsymbol{u}}_k(i)| = |\vec{\boldsymbol{\psi}}_k(i)|. \tag{15}$$

$\square$

## 1.3 Proving proposition 2

Given two graphs $G$ and $H$, let $\{\triangle_i^G\}_{i=1}^{n_G}$ and $\{\triangle_j^H\}_{j=1}^{n_H}$ be the respective set representations $\big(\triangle_i^G = (\vec{\boldsymbol{p}}_i, a_i^1, a_i^2, ..., a_i^L)$ and likewise $\triangle_j^H\big)$. Let $\boldsymbol{K}_{GG}$, $\boldsymbol{K}_{HH}$, and $\boldsymbol{K}_{GH}$ be the kernel matrices, induced by the embedding kernel $k = \otimes_{l=0}^{L} k_l$. That is, they are defined such that $(\boldsymbol{K}_{GG})_{ij} = k(\triangle_i^G, \triangle_j^G)$, $(\boldsymbol{K}_{HH})_{ij} = k(\triangle_i^H, \triangle_j^H)$, and $(\boldsymbol{K}_{GH})_{ij} = k(\triangle_i^G, \triangle_j^H)$.

---

**Proposition 2.** *Let $\mathcal{G}$ be the set of graphs with attribute domains $\mathcal{A}_1, \mathcal{A}_2, ..., \mathcal{A}_L$. Let $G$ and $H$ be two graphs in $\mathcal{G}$. Let $m_G$ and $m_H$ be the corresponding mean embeddings. Then the following functions are positive definite graph kernels defined on $\mathcal{G} \times \mathcal{G}$.*

$$K_1(G, H) = (c + \langle m_G, m_H \rangle_{\mathcal{H}})^d = (c + \frac{1}{n_G n_H} \vec{\mathbf{1}}_{n_G}^T \boldsymbol{K}_{GH} \vec{\mathbf{1}}_{n_H})^d, c > 0, d \in \mathbb{N}, \tag{16a}$$

$$K_2(G, H) = \exp(-\gamma \|m_G - m_H\|_{\mathcal{H}}^p) = \exp\big[-\gamma \mathrm{MMD}^p(\mu_G, \mu_H)\big], \gamma > 0, 0 < p \leq 2, \tag{16b}$$

*where* $\mathrm{MMD}(\mu_G, \mu_H) = (\frac{1}{n_G^2} \vec{\mathbf{1}}_{n_G}^T \boldsymbol{K}_{GG} \vec{\mathbf{1}}_{n_G} + \frac{1}{n_H^2} \vec{\mathbf{1}}_{n_H}^T \boldsymbol{K}_{HH} \vec{\mathbf{1}}_{n_H} - \frac{2}{n_G n_H} \vec{\mathbf{1}}_{n_G}^T \boldsymbol{K}_{GH} \vec{\mathbf{1}}_{n_H})^{\frac{1}{2}}$ *is the maximum mean discrepancy (MMD) [6].*

---

*Proof.* **(a).** We first consider two kernels $K_\alpha(G, H) = \langle m_G, m_H \rangle_{\mathcal{H}}$ and $K_\beta(G, K) = c$. It can be easily observed that $K_\alpha$ and $K_\beta$ are positive definite graph kernels. Since the sum and multiplication of positive definite kernels are still positive definite, we conclude that (16a) are positive definite.

**(b).** The positive definiteness of (16b) is obtained from Corollary 3 in [12]. $\square$

## 1.4 Proving theorem 2

---

**Theorem 2.** *Let $G$ and $H$ be two graphs with attribute domains $\mathcal{A}_1, ..., \mathcal{A}_L$. Let $\hat{\phi}_l : \mathcal{A}_l \to \mathbb{R}^{D_l}, l = 0, 1, ..., L$ be the approximate explicit feature maps. Let $\{(\vec{\boldsymbol{p}}_i, a_i^1, a_i^2, ..., a_i^L)\}_{i=1}^{n_G}$ and $\{(\vec{\boldsymbol{q}}_j, b_j^1, b_j^2, ..., b_j^L)\}_{j=1}^{n_H}$ be the respective set representations of $G$ and $H$. Then their approximate explicit graph embeddings, $\hat{m}_G$ and $\hat{m}_H$, are tensors in $\mathbb{R}^{D_0 \times D_1 \times ... \times D_L}$, and can be written as*

$$\hat{m}_G = \frac{1}{n_G} \sum_{i=1}^{n_G} \hat{\phi}_0(\vec{\boldsymbol{p}}_i) \circ \hat{\phi}_1(a_i^1) \circ ... \circ \hat{\phi}_L(a_i^L), \quad \hat{m}_H = \frac{1}{n_H} \sum_{j=1}^{n_H} \hat{\phi}_0(\vec{\boldsymbol{q}}_j) \circ \hat{\phi}_1(b_j^1) \circ ... \circ \hat{\phi}_L(b_j^L). \tag{17}$$

*That is, as $D_0, D_1, ..., D_L \to \infty$, we have $\langle \hat{m}_G, \hat{m}_H \rangle_{\mathcal{T}} \to \langle m_G, m_H \rangle_{\mathcal{H}}$.*

---

*Before we prove theorem 2, we first introduce a lemma about the inner product of multi-dimensional tensors.*

**Lemma 3.** *Let $U = \vec{u}^{(0)} \circ \vec{u}^{(1)} \circ ... \circ \vec{u}^{(L)}$ and $V = \vec{v}^{(0)} \circ \vec{v}^{(1)} \circ ... \circ \vec{v}^{(L)}$ be two rank-one tensors in $\mathbb{R}^{D_0 \times D_1 \times ... \times D_L}$. Then we have $\langle U, V \rangle_{\mathcal{T}} = \langle \vec{u}^{(0)}, \vec{v}^{(0)} \rangle \langle \vec{u}^{(1)}, \vec{v}^{(1)} \rangle ... \langle \vec{u}^{(L)}, \vec{v}^{(L)} \rangle$.*

*Proof.*

$$
\begin{aligned}
&\langle U, V \rangle_{\mathcal{T}} \\
&= \sum_{i_0=1}^{D_0} \sum_{i_1=1}^{D_1} ... \sum_{i_L=1}^{D_L} U_{i_1 i_2,...,i_L} V_{i_1 i_2,...,i_L} \\
&= \sum_{i_0=1}^{D_0} \sum_{i_1=1}^{D_1} ... \sum_{i_L=1}^{D_L} \vec{u}_{i_0}^{(0)} \vec{u}_{i_1}^{(1)} ... \vec{u}_{i_L}^{(L)} \vec{v}_{i_0}^{(0)} \vec{v}_{i_1}^{(1)} ... \vec{v}_{i_L}^{(L)} \\
&= \left( \sum_{i_0=1}^{D_0} \vec{u}_{i_0}^{(0)} \vec{v}_{i_0}^{(0)} \right) \left( \sum_{i_1=1}^{D_1} \vec{u}_{i_1}^{(1)} \vec{v}_{i_1}^{(1)} \right) ... \left( \sum_{i_L=1}^{D_L} \vec{u}_{i_L}^{(L)} \vec{v}_{i_L}^{(L)} \right) \\
&= \langle \vec{u}^{(0)}, \vec{v}^{(0)} \rangle \langle \vec{u}^{(1)}, \vec{v}^{(1)} \rangle ... \langle \vec{u}^{(L)}, \vec{v}^{(L)} \rangle.
\end{aligned}
\tag{18}
$$

$\square$

*Now we prove theorem 2.*

*Proof.* First we calculate $\langle \hat{m}_G, \hat{m}_H \rangle_{\mathcal{T}}$.

$$
\begin{aligned}
&\langle \hat{m}_G, \hat{m}_H \rangle_{\mathcal{T}} \\
&= \left\langle \frac{1}{n_G} \sum_{i=1}^{n_G} \hat{\phi}_0(\vec{p}_i) \circ \hat{\phi}_1(a_i^1) \circ ... \circ \hat{\phi}_L(a_i^L), \frac{1}{n_H} \sum_{j=1}^{n_H} \hat{\phi}_0(\vec{q}_j) \circ \hat{\phi}_1(b_j^1) \circ ... \circ \hat{\phi}_L(b_j^L) \right\rangle_{\mathcal{T}} \\
&= \frac{1}{n_G n_H} \sum_{i=1}^{n_G} \sum_{j=1}^{n_H} \left\langle \hat{\phi}_0(\vec{p}_i) \circ \hat{\phi}_1(a_i^1) \circ ... \circ \hat{\phi}_L(a_i^L), \hat{\phi}_0(\vec{q}_j) \circ \hat{\phi}_1(b_j^1) \circ ... \circ \hat{\phi}_L(b_j^L) \right\rangle_{\mathcal{T}} \\
&= \frac{1}{n_G n_H} \sum_{i=1}^{n_G} \sum_{j=1}^{n_H} \langle \hat{\phi}_0(\vec{p}_i), \hat{\phi}_0(\vec{q}_j) \rangle \langle \hat{\phi}_1(a_i^1), \hat{\phi}_1(b_j^1) \rangle ... \langle \hat{\phi}_1(a_i^L), \hat{\phi}_1(b_j^L) \rangle \\
&= \frac{1}{n_G n_H} \sum_{i=1}^{n_G} \sum_{j=1}^{n_H} \hat{k}_0(\vec{p}_i, \vec{q}_j) \hat{k}_1(a_i^1, b_j^1) ... \hat{k}_L(a_i^L, b_j^L),
\end{aligned}
\tag{19}
$$

where the 3rd equality holds because of lemma 3.

Next we calculate $\langle m_G, m_H \rangle_{\mathcal{H}}$.

$$
\begin{aligned}
&\langle m_G, m_H \rangle_{\mathcal{H}} \\
&= \left\langle \frac{1}{n_G} \sum_{i=1}^{n_G} \phi(\boldsymbol{p}_i, a_i^1, ..., a_i^L), \frac{1}{n_H} \sum_{j=1}^{n_H} \phi(\boldsymbol{q}_j, b_j^1, ..., b_j^L) \right\rangle_{\mathcal{H}} \\
&= \frac{1}{n_G n_H} \sum_{i=1}^{n_G} \sum_{j=1}^{n_H} \langle \phi(\boldsymbol{p}_i, a_i^1, ..., a_i^L), \phi(\boldsymbol{q}_j, b_j^1, ..., b_j^L) \rangle_{\mathcal{H}} \\
&= \frac{1}{n_G n_H} \sum_{i=1}^{n_G} \sum_{j=1}^{n_H} k\left[ (\boldsymbol{p}_i, a_i^1, ..., a_i^L), (\boldsymbol{q}_j, b_j^1, ..., b_j^L) \right] \\
&= \frac{1}{n_G n_H} \sum_{i=1}^{n_G} \sum_{j=1}^{n_H} k_0(\vec{p}_i, \vec{q}_j) k_1(a_i^1, b_j^1) ... k_L(a_i^L, b_j^L),
\end{aligned}
\tag{20}
$$

where the last equality holds because of the definition of the embedding kernel $k = \otimes_{l=0}^{L} k_l$. Since $\hat{k}_0(\vec{p}_i, \vec{q}_j) \rightarrow k_0(\vec{p}_i, \vec{q}_j)$, $\hat{k}_1(a_i^1, b_j^1) \rightarrow k_1(a_i^1, b_j^1)$,..., $\hat{k}_L(a_i^L, b_j^L) \rightarrow k_L(a_i^L, b_j^L)$, as $D_0, D_1, ..., D_L \rightarrow \infty$, we conclude that $\langle \hat{m}_G, \hat{m}_H \rangle_{\mathcal{T}} \rightarrow \langle m_G, m_H \rangle_{\mathcal{H}}$. $\square$

## 1.5 Proving proposition 3

**Proposition 3.** *Let $\mathcal{G}$ be the set of graphs with attribute domains $\mathcal{A}_1, \mathcal{A}_2, ..., \mathcal{A}_L$. The following functions are positive definite graph kernels defined on $\mathcal{G} \times \mathcal{G}$.*

$$\hat{K}_1(G, H) = (c + \langle \hat{m}_G, \hat{m}_H \rangle_\mathcal{T})^d = \left[ c + \text{vec}(\hat{m}_\mathcal{G})^T \text{vec}(\hat{m}_\mathcal{H}) \right]^d, c > 0, d \in \mathbb{N}$$

$$\hat{K}_2(G, H) = \exp(-\gamma \| \hat{m}_G - \hat{m}_H \|_\mathcal{T}^p) = \exp(-\gamma \| \text{vec}(\hat{m}_G) - \text{vec}(\hat{m}_H) \|_2^p), \gamma > 0, 0 < p \le 2.$$

*As $D_0, D_1, ..., D_L \to \infty$, we have $\hat{K}_1(G, H) \to K_1(G, H)$ and $\hat{K}_2(G, H) \to K_2(G, H)$.*

*Proof.* The positive definiteness of $\hat{K}_1$ and $\hat{K}_2$ can be proved in the same way with Theorem 2. The convergence property can be obtained by Theorem 2. $\square$

## 2 Datasets description

The statistics of the benchmark graph datasets used in the paper are reported in Table 1. Next, we describe in these datasets in detail.

### 2.1 Non-attributed (unlabeled) graph datasets

**COLLAB** [15] is a scientific collaboration dataset that consists of the ego-networks of 5,000 researchers from three scientific fields: *High Energy Physics*, *Condensed Matter Physics*, and *Astro Physics*. The task is to determine the field of each researcher based on their ego-networks.

**IMDB-BINARY** [15] is a movie collaboration dataset that consists of the ego-networks of 1,000 actors/actresses who played roles in movies in IMDB. In each graph, nodes represent actors/actress, and there is an edge between them if they appear in the same movie. These graphs are derived from the *Action* and *Romance genres*.

**IMDB-MULTI** [15] is generated in a similar way to **IMDB-BINARY**. The difference is that it is derived from three genres: *Comedy*, *Romance*, and *Sci-Fi*.

**REDDIT-BINARY** [15] consists of graphs corresponding to online discussions on Reddit. In each graph, nodes represent users, and there is an edge between them if at least one of them respond to the other's comment. There are four popular subreddits, namely, *IAmA*, *AskReddit*, *TrollXChromosomes*, and *atheism*. *IAmA* and *AskReddit* are two question/answerbased *subreddits*, and *TrollXChromosomes* and *atheism* are two discussion-based subreddits. A graph is labeled according to whether it belongs to a question/answer-based community or a discussion-based community.

**REDDIT-MULTI(5K)** [15] is generated in a similar way to **REDDIT-BINARY**. The difference is that there are five subreddits involved, namely, *worldnews*, *videos*, *AdviceAnimals*, *aww*, and *mildlyinteresting*. Graphs are labeled with their corresponding subreddits.

**REDDIT-MULTI(12K)** [15] is generated in a similar way to **REDDIT-BINARY** and **REDDIT-MULTI(5K)**. The difference is that there are eleven subreddits involved, namely, *AskReddit*, *AdviceAnimals*, *atheism*, *aww*, *IAmA*, *mildlyinteresting*, *Showerthoughts*, *videos*, *todayilearned*, *worldnews*, and *TrollXChromosomes*. Still, graphs are labeled with their corresponding subreddits.

### 2.2 Graphs with discrete attributes

**MUTAG** [3] consists of graph representations of 188 mutagenic aromatic and heteroaromatic nitro chemical compounds. These graphs are labeled according to whether or not they have a mutagenic effect on the Gramnegative bacterium Salmonella typhimurium.

**DD** [4] consists of graph representations of 1,178 proteins. In each graph, nodes represent amino acids, and there is an edge if they are less than six Angstroms apart. Graphs are labeled according to whether they are enzymes or not.

**NCI1** [13] consists of graph representations of 4,110 chemical compounds s screened for activity against non-small cell lung cancer and ovarian cancer cell lines, respectively.

**PTC** [7] consists of graph representations of chemical molecules. In each graph, nodes represent atoms, and edges represent chemical bonds. Graphs are labeled according to carcinogenicity on rodents, divided into male mice (**MM**), male rats (**MR**), female mice (**FM**), and female rats (**FR**).

### 2.3 Graphs with continuous attributes

**FRANK** [8] is a chemical molecule dataset that consists of 2,401 mutagens and 1,936 nonmutagens. Originally, nodes are associated with chemical atom symbols. The most frequent atom symbols are mapped to MNIST digit images. By doing this, the original atom symbols can be recovered through the high dimensional MNIST vectors of pixel intensities, which are treated as the continuous attributes on graphs.

**SYNTHETIC** [5] consists of 300 random graphs. The continuous node attributes are sampled from the distribution $N(0, 1)$. There are two classes, A and B. Class A has 150 graphs, which are generated by randomly rewiring five edges and permuting ten node attributes. Class B has 150 graphs, which are generated by randomly rewiring ten edges and permuting five node attributes.

**Synthie** [10] consists of 400 random graphs, all of which are variants of two Erdos-Renyi graphs. The nodes are associated with 15-dimensional continuous attributes. All graphs are divided into four classes. The generation process of these graphs is described in [10].

### 2.4 Graphs with both discrete and continuous attributes

**ENZYMES** and **PROTEINS** [1] consist of graph representations of proteins. Nodes represent secondary structure elements (SSE), and there is an edge if they are neighbors along the amino acid sequence or one of three neareset neighbors in space. The discrete attributes are SSE's types. The continuous attributes are the 3D length of the SSE. Graphs are labeled according to which EC top-level class they belong to.

**BZR**, **COX2**, and **DHFR** [14], [9] all are chemical compound datasets. Still, in each graph, nodes represent atoms, and edges represent chemical bonds. The discrete attributes correspond to atom types. The continuous attributes are 3D coordinates.

Table 1: Statistics of the benchmark graph datasets

| Datasets | graph # | class # | average node # | average edge # | discrete attributes | continuous attributes (Dim) |
|---|---|---|---|---|---|---|
| COLLAB | 5000 | 3 | 74.49 | 2457.78 | × | × |
| IMDB-BINARY | 1000 | 2 | 19.77 | 96.53 | × | × |
| IMDB-MULTI | 1500 | 3 | 13.00 | 65.94 | × | × |
| REDDIT-BINARY | 2000 | 2 | 429.63 | 497.75 | × | × |
| REDDIT-MULTI(5K) | 4999 | 5 | 508.52 | 594.87 | × | × |
| REDDIT-MULTI(12K) | 11929 | 11 | 391.41 | 456.89 | × | × |
| MUTAG | 188 | 2 | 17.93 | 19.79 | √ | × |
| DD | 1178 | 2 | 284.32 | 715.66 | √ | × |
| NCI1 | 4110 | 2 | 29.87 | 32.30 | √ | × |
| PTC-FM | 349 | 2 | 14.11 | 14.48 | √ | × |
| PTC-FR | 351 | 2 | 14.56 | 15.00 | √ | × |
| PTC-MM | 336 | 2 | 13.97 | 14.32 | √ | × |
| PTC-MR | 344 | 2 | 14.29 | 14.69 | √ | × |
| FRANK | 4337 | 2 | 16.90 | 17.88 | × | √ (780) |
| SYNTHETIC | 300 | 2 | 100 | 196.25 | × | √ (1) |
| Synthie | 400 | 4 | 95.00 | 172.93 | × | √ (15) |
| ENZYMES | 600 | 6 | 32.63 | 64.14 | √ | √ (18) |
| PROTEINS | 1113 | 2 | 39.06 | 72.82 | √ | √ (1) |
| BZR | 405 | 2 | 35.75 | 38.36 | √ | √ (3) |
| COX2 | 467 | 2 | 41.22 | 43.45 | √ | √ (3) |
| DHFR | 467 | 2 | 42.43 | 44.54 | √ | √ (3) |

## 3 Additional Experiments

### 3.1 Experimental results of linear/nonlinear transformation on top of graph kernels

In Table 2, we empirically show that the observed improvements are mainly due to the power of our RPF and Hilbert space embeddings, instead of the nonlinear transformations on top of RPF and

node information. We show the classification results of our $\mathrm{RetGK_{II}}$ with linear transformation, the modified WL and HGK kernels with RBF transformation on the top of WL and HGK. Comparing the results in Table 2 with those shown in the paper, we can see that: (1) with linear transformation, $\mathrm{RetGK_{II}}$ still keeps the superior performance; (2) with RBF transformation, WL and HGK don't get observable improvements (in some cases, their performance even degrades).

| Data | | COLLAB | IMDB(B) | IMDB(M) | | REDDIT(B) | REDDIT(M5K) | | REDDIT(M12K) |
|---|---|---|---|---|---|---|---|---|---|
| $\mathrm{RetGK_{II}}$ | | 78.6(0.4) | 72.3(0.6) | 48.3(0.6) | | 89.9(0.4) | 56.3(0.3) | | 48.0(0.2) |
| Data | ENZYMES | PROTEIN | MUTAG | D&D | NCl1 | PTC(FM) | PTC(FR) | PTC(MM) | PTC(MR) |
| WL | 55.3(1.2) | 72.0(0.6) | 84.6(1.5) | 78.6(0.4) | 85.2(0.3) | 56.1(2.2) | 62.7(1.9) | 60.2(1.9) | 55.6(1.7) |
| Data | | ENZYMES | | PROTEIN | | FRANK | SYNTHETIC | | Synthie |
| HGK | | 60.7(1.1) | | 74.7(0.7) | | 66.7(0.3) | 94.7(1.0) | | 84.5(0.4) |

Table 2: The classification results of $\mathrm{RetGK_{II}}$ (with linear transformation), WL (with RBF transformation), and HGK (with RBF transformation), respectively. Top two rows: unlabeled. Middle two rows: discrete labels only. Bottom two rows: continuous labels only

## 3.2 Experimental results of graph kernels using eigenvector embeddings

In Table 3, we show the experimental results of using eigenvector embeddings to obtain approximated Hilbert embeddings. The experiment setup is the same with that in the paper. The embedding dimension $d$ is selected from $\{2, 3, 4, 5, 10, 15, 20\}$. Clearly, RPF significantly outperforms EE.

| COLLAB | | IMDB(B) | | IMDB(M) | | REDDIT(B) | | REDDIT(M5K) | | REDDIT(M12K) |
|---|---|---|---|---|---|---|---|---|---|---|
| 78.1(0.3) | | 71.6(0.4) | | 43.2(0.3) | | 88.7(0.4) | | 50.0(0.5) | | 40.4(0.3) |
| ENZYMES | PROTEIN | MUTAG | D&D | NCl1 | PTC(FM) | PTC(FR) | PTC(MM) | PTC(MR) | | |
| 46.2(1.3) | 72.7(0.5) | 87.0(1.1) | 79.6(0.5) | 81.5(0.3) | 61.7(1.3) | 62.4(1.4) | 60.8(1.6) | 61.1(1.7) | | |
| ENZYMES | PROTEIN | FRANK | SYNTHETIC | Synthie | ENZYMES | PROTEIN | BZR | COX2 | | DHFR |
| 69.4(1.3) | 72.3(0.8) | 72.1(0.4) | 83.5(0.9) | 44.0(3.1) | 69.5(0.8) | 75.1(0.5) | 80.1(0.8) | 77.1(1.1) | | 81.5(0.6) |

Table 3: The classification results of eigenvector embeddings. Top two rows: unlabeled. Middle two rows: discrete labels only. Bottom two rows: continuous labels only (left), discrete and continuous labels (right).