[Reviews · NeurIPS 2018]

Reviewer 1



The paper proposes a kernel for graphs able to deal with discrete and continuous labels. In particular, the topology information of a graph is encoded at node level by a return random walk probability vector (each dimension being associated to a different walk length). This probability vector is obtained by classical equations used by random walk kernels for graphs (T. Gartner et al. classical LTKM paper) but just focussing on return random walks. Thanks to that the computational complexity can be reduced since only the entries on the diagonal of the powers of the transition probability matrix need to be computed. This can be done via eigen-decomposition of a rescaled version of the adjacency matrix. Notwithstanding that, for large graphs the computational complexity is still too high and Monte Carlo methods are suggested as a practical way to get estimates of the quantities of interest. Having obtained the (approximated) return probability vector for each node of the graph, an embedding of a graph combining also the original information attached to each node (via tensor product of kernels) is obtained by kernel mean embedding (proposed in a JMLR 2012 paper). Again, the embedding is computationally too expensive for large graphs, so the paper proposes to exploit previous work on approximate explicit feature maps (NIPS 2008 paper by Rahimi & Recht) to get a moderate size explicit tensor representation of graphs. All the proposed kernels exploit either a polynomial or a RBF transformation (exploiting random Fourier feature map for the approximated version). Experimental assessment is performed on classical benchmarking datasets involving just structural information or additional discrete and continuous information at node level. Quality: The mathematical treatment is solid as well as all the concepts that are used as building blocks of the proposed approach. I just have few concerns on the following points: i) one of the problems of random walks is tottering (Mahe et al., ICML 2004), i.e. the fact that short paths in the graph can be visited many times forward and backward by a walk, thus over representing that topological feature with respect to walks supported by longer paths. This problem is particularly severe for return walks. In the paper there is no discussion about this well known problem as well as there is no analysis of the impact of this problem on the quality of the obtained vectorial representations. ii) the properties of the return probability features (RPF) may not be preserved by their approximated versions. For example, isomorphism-invariance is not valid anymore since the same graph with different (or even same) representations can get different RPFs because of the Monte Carlo approximation. The paper does not mention this issues as well as does not present empirical evidence of the quality of the obtained representations. iii) since nonlinear transformations (polynomial and RBF) are used on top of RPFs and additional node information, it is not clear to me how fair is the experimental comparison. In fact, I guess that a fair comparison would imply the use of the very same kernels on top of the other kernels taken from the literature. It is not clear to me how much of the observed improvements (in some cases marginal with respect to the state-of-the-art) are due to that. iv) concerning the experimental comparison versus graph kernels for continuous attributes, the author missed a recent proposal that is relevant and uses some of the paper datasets: Giovanni Da San Martino, Nicolò Navarin, Alessandro Sperduti: Tree-Based Kernel for Graphs With Continuous Attributes. IEEE Trans. Neural Netw. Learning Syst. 29(7): 3270-3276 (2018). Clarity: the paper is well written and can be followed with no problem by an expert in the field. Organisation of the material is good. The level of presentation allows the reader to both have a general idea of the flow of reasoning and some technical details that characterise the proposed approach. Originality: the paper builds on results from the literature. The idea to just use the return probabilities of random walks is new to my knowledge, although it can be considered a bit derivative. Some effort has been put into the derivation of the theoretical results concerning RPFs and new proposed kernels. Overall I see some original contribution here. Significance: I guess the main contribution of the proposed kernels is the possibility to deal with a large variety of graphs (with no information/discrete/continuous attached to nodes). Not sure about the impact on cases where information is attached to nodes. Follow-up could involve other ways to code structural information. After rebuttal. I am not convinced about the argument with tottering. Even if labels are not considered, still there will be a strong bias on closer nodes. The authors should find a more convincing argument.

Reviewer 2



The paper presents a novel graph kernel, adapted to large undirected graphs, able to use nodes attributes. The idea is to use a "return probability feature" based on random walks, instead of classical bags of random walks for instance. The idea is interesting and appears to be efficient. The paper is quite easy to read but some details can be improved (see below). To my knowledge, the proposed approach is novel, and there are many potential problems that could benefit from this idea. It is shown that the RPF has several nice properties, such as isomorphic invariance and multi-resolution. It also contains enough information on graph structure such that similar RPF is linked to similar spectral properties. Details: *l102-103 : a little short I think *133 : times : \times? *eq(5) and (6) : nice presentation * 3.2.1 : would Nystrom methods be able to help here? * eq (7) : not clear * experiments : poly kernel not used? why presetting it then? or why not using it? * fig 2 : missing axis legend * biblio [17] : published in ICML 2012 * bibio [31] : missing venue (AAAI)

Reviewer 3



The authors devise graph kernels based on return probabilities of random walks. They prove that this captures the spectral information of the graphs. Based on the return probabilities, Hilbert space embeddings for graphs with discrete and continuous attributes are derived, which induce valid kernels. Moreover, two ways to speed up the kernel computation are proposed: (i) sampling random walks and (ii) using Random Fourier features. The main strengths of the paper are: * Seemingly new idea how to use random walks to derive graph kernels. * The approach is well motivated and justified. I like the toy example in Figure 1! * The paper is well-written and clear. * Experimental comparison on many data sets. The weak points of the paper are: * It is not clear to me why the proposed return probabilities should perform better than directly using the information provided by the eigenvectors of the adjacency matrix. The relation proven seems natural. The eigenvectors are for example used in this publication to obtain node embeddings: "Matching Node Embeddings for Graph Similarity", Giannis Nikolentzos, Polykarpos Meladianos, Michalis Vazirgiannis, AAAI 2017 I would like to propose to use the eigenvector embedding of nodes (just as the return probabilities) to obtain a Hilbert space embedding and use these as a baseline for comparison. * The accuracy values in the experimental evaluation give rise to several questions: 1) The tables contain a kernel "CSM", which is not explained in the text. I guess that this is the "Connected Subgraph Matching kernel" (Ref 17). The accuracy values seem to be copied from that reference. 2) Which values are obtained based on your own experiments and which are copied from the literature? Comparing against the values from literature is often not fair, since the data sets may differ (e.g., edge labels are ignored or taken into account, only a subset of the attributes are considered etc.) and test splits are different. * Other state-of-the-art competitors for comparison would be the Weisfeiler-Lehman optimal assignment kernel and the Multiscale Laplacian kernel (both NIPS 2016). Remarks: * It would be interesting to provide bounds on the number of random walks needed to approximate the exact kernel with a certain quality. * Simply summing up the running time over all data sets is not very insightful. * Table 3: HGK-SP is not considered. * Table 4: HGK can also handle discrete+cont. attributes (the numbers are actually provided in the corresponding paper). * p2, l64: 'permutataion map' should only be used when domain and codomain coincide. In summary, I can only recommend the acceptance of the paper, provided that (i) the difference to the natural eigenvector embedding is discussed adequately and (ii) the issues regarding the experimental evaluation have been sorted out. ========================= Update after the rebuttal: My concerns about (i) have been addressed. Regarding (ii) I would like to suggest to make clear, which values were copied from the literature and which were obtained from your own experiments. For the values copied from the literature, please make sure that they are obtained for the same data set. This is, for example, not the case for CSM on ENZYMES, since in the experiments in Ref 17 only a subset of the available attributes were used.